# Combining Denoising Autoencoders with Contrastive Learning to fine-tune Transformer Models

**Alejo López-Ávila**
Huawei London Research Centre
London, UK

alejo.lopez.avila@huawei.com

**Víctor Suárez-Paniagua**
Huawei Ireland Research Center
Dublin, Ireland

victor.suarez.paniagua@huawei-partners.com

## Abstract

Recently, using large pre-trained Transformer models for transfer learning tasks has evolved to the point where they have become one of the flagship trends in the Natural Language Processing (NLP) community, giving rise to various outlooks such as prompt-based, adapters, or combinations with unsupervised approaches, among many others. In this work, we propose a $3\text{-}Phase$ technique to adjust a base model for a classification task. First, we adapt the model's signal to the data distribution by performing further training with a Denoising Autoencoder (*DAE*). Second, we adjust the representation space of the output to the corresponding classes by clustering through a Contrastive Learning (*CL*) method. In addition, we introduce a new data augmentation approach for Supervised Contrastive Learning to correct the unbalanced datasets. Third, we apply fine-tuning to delimit the predefined categories. These different phases provide relevant and complementary knowledge to the model to learn the final task. We supply extensive experimental results on several datasets to demonstrate these claims. Moreover, we include an ablation study and compare the proposed method against other ways of combining these techniques.

## 1 Introduction

The never-ending starvation of pre-trained Transformer models has led to fine-tuning these models becoming the most common way to solve target tasks. A standard methodology uses a pre-trained model with self-supervised learning on large data as a base model. Then, replace/increase the deep layers to *learn* the task in a supervised way by leveraging knowledge from the initial base model. Even though specialised Transformers, such as Pegasus (Zhang et al., 2020) for summarisation, have appeared in recent years, the complexity and resources needed to train Transformer models of these scales make fine-tuning methods the most

reasonable choice. This paradigm has raised interest in improving these techniques, either from the point of the architecture (Qin et al., 2019), by altering the definition of the fine-tuning task (Wang et al., 2021b) or the input itself (Brown et al., 2020), but also by revisiting and proposing different self-supervised training practices (Gao et al., 2021). We decided to explore the latter and offer a novel approach that could be utilised broadly for NLP classification tasks. We combine some self-supervised methods with fine-tuning to prepare the base model for the data and the task. Thus, we will have a model adjusted to the data even before we start fine-tuning without needing to train a model from scratch, thus producing better results, as shown in the experiments.

A typical way to adapt a neural network to the input distribution is based on Autoencoders. These systems introduce a bottleneck for the input data distribution through a reduced layer, a sparse layer activation, or a restrictive loss, forcing the model to reduce a sample's representation at the narrowing. A more robust variant of this architecture is *DAE*, which corrupts the input data to prevent it from learning the identity function. The first phase of the proposed method is a *DAE* (Fig. 1a), replacing the final layer of the encoder with one more adapted to the data distribution.

Contrastive Learning has attracted the attention of the NLP community in recent years. This family of approaches is based on comparing an anchor sample to negative and positive samples. There are several losses in *CL*, like the triplet loss (Schroff et al., 2015), the contrastive loss (Chopra et al., 2005), or the cosine similarity loss. The second phase proposed in this work consists of a Contrastive Learning using the cosine similarity (as shown in Fig. 1b). Since the data is labelled, we use a supervised approach similar to the one presented in (Khosla et al., 2020) but through Siamese Neural Networks. In contrast, we consider Contrastive

Learning and fine-tuning the classifier (*FT*) as two distinct stages. We also add a new imbalance correction during data augmentation that avoids overfitting. This *CL* stage has a clustering impact since the vector representation belonging to the same class will tend to get closer during the training. We chose some benchmark datasets in classification tasks to support our claims. For the hierarchical ones, we can adjust labels based on the number of similar levels among samples.

Finally, once we have adapted the model to the data distribution in the first phase and clustered the representations in the second, we apply fine-tuning at the very last. Among the different variants from fine-tuning, we use the traditional one for Natural language understanding (NLU), i.e., we add a small Feedforward Neural Network (FNN) as the classifier on top of the encoder with two layers. We use only the target task data without any auxiliary dataset, making our outlook self-contained. The source code is publicly available at GitHub[1].

To summarise, our contribution is fourfold:

1. We propose a $3\text{-}Phase$ fine-tuning approach to adapt a pre-trained base model to a supervised classification task, yielding more favourable results than classical fine-tuning.

2. We propose an imbalance correction method by sampling noised examples during the augmentation, which supports the Contrastive Learning approach and produces better vector representations.

3. We analyze possible ways of applying the described phases, including ablation and joint loss studies.

4. We perform experiments on several well-known datasets with different classification tasks to prove the effectiveness of our proposed methodology.

## 2  Related Work

One of the first implementations was presented in (Reimers and Gurevych, 2019), an application of Siamese Neural Networks using BERT (Devlin et al., 2018) to learn the similarity between sentences.

Autoencoders were introduced in (Kramer, 1991) and have been a standard for self-supervised learning in Neural Networks since then. However, new modifications were created with the explosion of Deep Learning architectures, such as *DAE* (Vincent et al., 2010) and masked Autoencoders (Germain et al., 2015). The Variational Autoencoder (*VAE*) has been applied for NLP in (Miao et al., 2015) or (Li et al., 2019) with RNN networks or a *DAE* with Transformers in (Wang et al., 2021a). Transformers-based Sequential Denoising Auto-Encoder (Wang et al., 2021a) is an unsupervised method for encoding the sentence into a vector representation with limited or no labelled data, creating noise through an MLM task. The authors of that work evaluated their approach on the following three tasks: Information Retrieval, Re-Ranking, and Paraphrase Identification, showing an increase of up to $6.4$ points compared with previous state-of-the-art approaches. In (Savinov et al., 2021), the authors employed a new Transformer architecture called Step-unrolled Denoising Autoencoders. In the present work, we will apply a *DAE* approach to some Transformer models and extend its application to sentence-based and more general classification tasks.

The first work published on Contrastive Learning was (Chopra et al., 2005). After that, several versions have been created, like the triplet net in Facenet (Schroff et al., 2015) for Computer Vision. The triplet-loss compares a given sample and randomly selected negative and positive samples making the distances larger and shorter, respectively. One alternative approach for creating positive pairs is slightly altering the original sample. This method was followed by improved losses such as $N\text{-}pair\,Loss$ (Sohn, 2016) and the Noise Contrastive Estimation ($NCE$) (Gutmann and Hyvärinen, 2010), extending the family of *CL* techniques. In recent years, further research has been done on applying these losses, e.g. by supervised methods such as (Khosla et al., 2020), which is the one that most closely resembles one in our second phase.

Sentence-BERT (Reimers and Gurevych, 2019) employs a Siamese Neural Network using BERT with a pooling layer to encode two sentences into a sentence embedding and measure their similarity score. Sentence-BERT was evaluated on Semantic Textual Similarity ($STS$) tasks and the SentEval toolkit (Conneau and Kiela, 2018), outperforming other embedding strategies in most tasks. In

---

[1] https://github.com/vsuarezpaniagua/3-phase_finetuning

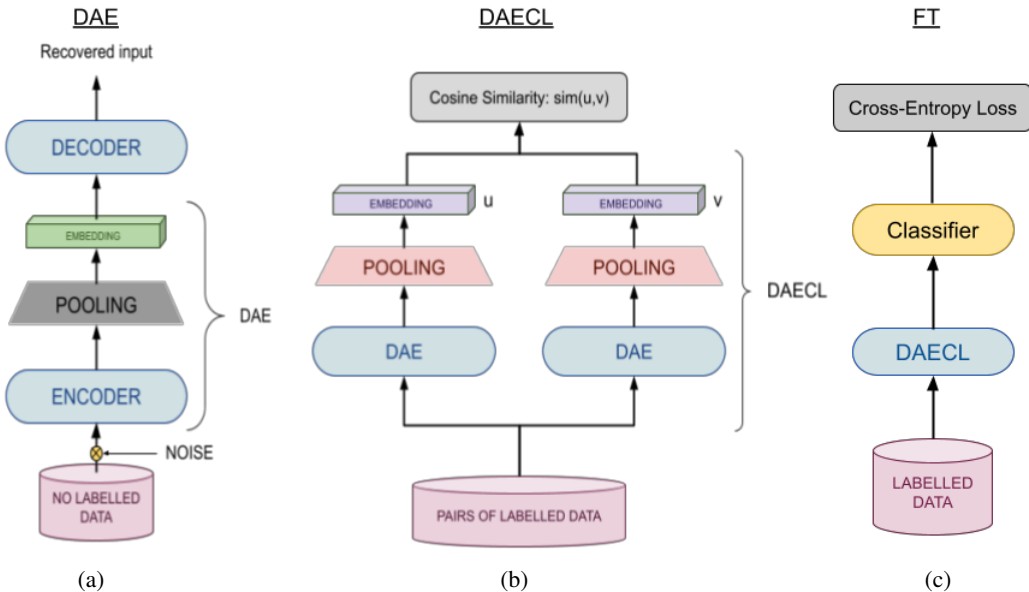

Figure 1: (a) First stage: Denoising Autoencoder architecture where the encoder and the decoder are based on the pre-trained Transformers. The middle layer following the pooling together with the encoder model will be the resulting model of this stage. (b) Second stage: Supervised Contrastive Learning phase. We add a pooling layer to the previous one to learn the new clustered representation. The set of blocks denoted as *DAECL* will be employed and adapted as a base model in the fine-tuning phase. (c) Third stage: Classification phase through fine-tuning

our particular case, we also use Siamese Networks within the *CL* options. Similar to this approach, the Simple Contrastive Sentence Embedding (Gao et al., 2021) is used to produce better embedding representations. This unlabelled data outlook uses two different representations from the same sample, simply adding the noise through the standard dropout. In addition, they tested it using entailment sentences as positive examples and contradiction sentences as negative examples and obtained better results than $SBERT$ in the $STS$ tasks.

Whereas until a few years ago, models were trained for a target task, the emergence of pre-trained Transformers has changed the picture. Most implementations apply transfer learning on a Transformer model previously pre-trained on general NLU, on one or more languages, to the particular datasets and for a predefined task.

This new paradigm has guided the search for the best practice in each of the *trending areas* such as prompting strategies (Brown et al., 2020), Few Shot Learning (Wang et al., 2021b), meta-learning (Finn et al., 2017), also some concrete tasks like Intent Detection (Qin et al., 2019), or noisy data (Siddhant Garg, 2021). Here, we present a task-agnostic approach, outperforming some competitive methods for their corresponding tasks. It should also be noted that our method benefits from

using mostly unlabelled data despite some little labelled data. In many state-of-the-art procedures, like (Sun et al., 2020a), other datasets than the target dataset are used during training. In our case, we use only the target dataset.

## 3 Model

In this section, we describe the different proposed phases: a Denoising Autoencoder that makes the inputs robust against noise, a Contrastive Learning approach to identify the similarities between different samples in the same class and the dissimilarities with the other class examples together with a novel imbalance correction, and finally, the traditional fine-tuning of the classification model. In the last part of the section, we describe another approach combining the first two phases into one loss.

### 3.1 *DAE*: Denoising Autoencoder phase

The Denoising Autoencoder is the first of the proposed $3\text{-}Phase$ approach, shown in Fig. 1a. Like any other Autoencoder, this model consists of an encoder and a decoder, connected through a bottleneck. We use two Transformer models as encoders and decoders simultaneously: $RoBERTa$ (Liu et al., 2019), and $all\text{-}MiniLM\text{-}L12\text{-}v2$ (Wang et al., 2020). The underlying idea is that the bottleneck represents the input according to the general

| Dataset | Train DAE | Train SCL | Test |
|---------|-----------|-----------|------|
| SST2    | 69170     | 67349     | 872  |
| SNIPS   | 262       | 262       | 65   |
| SNIPS2  | 13084     | 13084     | 700  |
| SST5    | 8544      | 8544      | 2210 |
| AGNews  | 120K      | 120K      | 7600 |
| IMDB    | 25K       | 25K       | 25K  |

Table 1: Statistics for Train, Validation and Test dataset splits.

| Dataset | Avg. Length | Max. Length |
|---------|-------------|-------------|
| SST2    | 9           | 52          |
| SNIPS   | 9           | 20          |
| SNIPS2  | 9           | 35          |
| SST5    | 19          | 56          |
| AGNews  | 37          | 177         |
| IMDB    | 229         | 2450        |

Table 2: Average and max lengths for each of the datasets mentioned in the paper.

distribution of the whole dataset. A balance needs to be found between preventing the information from being memorised and having sufficient sensitivity to be reconstructed by the decoder, forcing the bottleneck to learn the general distribution. We add noise to the Autoencoder to prevent the bottleneck from memorising the data. We apply a Dropout on the input to represent the noise. Formally, for a sequence of tokens $X = \{x_0, \cdots, x_n\}$ coming from a data distribution $D$, we define the loss as

$$\mathcal{L}_{DAE} = \mathbb{E}_D[\log P_\theta(X|\bar{X})] \tag{1}$$

where $\bar{X}$ is the sequence $X$ after adding the noise. To support with an example, masking a token at the position $i$ would produce $\bar{X} = \{x_0, \cdots, x_{i-1}, 0, x_{i+1}, \cdots, x_n\}$. The distribution of $P_\theta$ in this Cross-Entropy loss corresponds to the composition of the decoder with the encoder. We consider the ratio of noise like another hyperparameter to tune. More details can be found in Appendix A.1, and the results section 5. Instead of applying the noise to the dataset and running a few epochs over it, we apply the noise on the fly, getting a very low probability of a repeated input.

Once the model has been trained, we extract the encoder and the bottleneck, resulting in *DAE* (Fig. 1a), which will be the encoder for the next

step. Each model's hidden size has been chosen as its bottleneck size (768 in the case of $RoBERTa$). The key point of the first phase is to adapt the embedding representation of the encoder to the target dataset distribution, i.e., this step shifts the distribution from the general distribution learned by the pre-trained Transformers encoder into one of the target datasets. It should be noted here that this stage is in no way related to the categories for qualification. The first phase was implemented using $SBERT$ library.[2]

## 3.2   *CL*: Contrastive Learning phase

The second stage will employ Contrastive Learning, more precisely, a Siamese architecture with cosine similarity loss. The contrastive techniques are based on comparing pairs of examples or anchor-positive-negative triplets. Usually, these methods have been applied from a semi-supervised point of view. We decided on a supervised outlook where the labels to train in a supervised manner are the categories for classification, i.e., we pick a random example. Then, we can get a negative input by sampling from the other categories or a positive one by sampling from the same category. We combine this process of creating pairs with the imbalance correction explained below to get pairs of vector outputs $(u, v)$. Given two inputs $v$ and $u$ and a label $label_{u,v}$ based on their class similarity.

$$label_{u,v} = \begin{cases} 1, & \text{if } u \text{ and } v \text{ are in the same class.} \\ 0, & \text{otherwise.} \end{cases} \tag{2}$$

We use a straightforward extension for the hierarchical dataset ($AGNews$) by normalising these weights along the number of levels. In the case of two levels, we assign 1 for the case where all the labels match, $0.5$ when only the first one matches, and $0$ if none. After applying the encoder, we obtain $\bar{u}$ and $\bar{v}$. We define the loss over these outputs as the Mean Squared Error ($MSE$):

$$\mathcal{L}_{CL} = ||label_{u,v} - CosineSim(u, v))||_2 \tag{3}$$

We apply this Contrastive Learning to the encoder *DAE*, i.e., the encoder and the bottleneck from the previous step. Again, we add an extra layer to this encoder model, producing our following final embedding. We denote this encoder after applying *CL* over *DAE* as *DAECL* (Fig. 1b). Similarly, we chose the hidden size per model as the

---

[2]https://www.sbert.net/index.html

embedding size. *CL* plays the role of soft clustering for the embedding representation of the text based on the different classes. This will make fine-tuning much easier as it will be effortless to distinguish representations from distinct classes. The second phase was also implemented through the $SBERT$ library.

### 3.2.1 Imbalance correction

As mentioned in the previous section, we are using a Siamese network and creating the pairs in a supervised way. It is a common practice to augment the data before applying *CL*. In theory, we can make as many example combinations as possible. In the case of the Siamese models, we have a total of $n!$ unique pair combinations that correspond to the potential number of pairs based on symmetry. Typically, data can be increased as much as one considers suitable. The problem is that one may fall short or overdo it and produce overfitting in the smaller categories. Our augmentation is based on two pillars. We start with correcting the imbalance in the dataset by selecting underrepresented classes in the dataset more times but without repeating pairs. Secondly, on the classes that have been increased, we apply noise on them to provide variants that are close but not the same to define more clearly the cluster to which they belong.

We balance the dataset by defining the number of examples that we are going to use per class based on the most significant class (where $max_k$ is its size) and a range of ratios, from $\min_{ratio}$ the minimum and $max_{ratio}$ the maximum. These upper and lower bounds for the ratios are hyper-parameters, and we chose 4 and 1.5 as default values. Formally, the new ratio is defined by the function:

$$f(x) = \log\left(\frac{max_k}{x}\right) \times \frac{max_{ratio}}{\log max_k} \quad (4)$$

where $x$ refers to the initial size of a given class. After adding a lower bound for the ratio, we get the final amount

$$newratio_k = \min\left(\min_{ratio}, f(class_k)\right) \quad (5a)$$

$$newclass_k = newratio_k \times class_k \quad (5b)$$

for $k = 1, 2, \cdots, K$ in a classification problem with $K$ classes, where $class_k$ and $newclass_k$ are the cardinalities of the class $k$, before and after the resizing, respectively. As we can observe, the

function 4 gives $f(1) = max_{ratio}$ for one example, so we will never get something bigger than the maximum. The shape of this function is similar to a negative log-likelihood, given a high ratio to the small classes and around the minimum for medium or bigger. A simple computation shows that the ratio 1 in function 4 is obtained at

$$x = max_k^{(max_{ratio}-1)/max_{ratio}} \quad (6)$$

or $max_k^{0.75}$ in our default case.

We duplicate this balanced dataset and shuffle both to create the pairs. Since many combinations exist, we only take the unique ones without repeating pairs, broadly preserving the proportions. Even so, if just a few examples represent one class, the clustering process from *CL* can be affected by the augmentation because the border between them would be defined just for a few examples. To avoid this problem, we add a slight noise to the tokens when the text length is long enough. This noise consists of deleting some of the stop-words from the example. Usually, this noise was added to create positive samples and produced some distortion to the vector representation. Adding it twice, in the augmentation and the *CL*, would produce too much distortion. However, since we are using a supervised approach, this does not negatively affect the model, as shown in the ablation section 5.

### 3.3 FT: Fine-tuning phase

Our final stage is fine-tuning, obtained by employing *DAECL* as our base model (as indicated in Fig 1b). We add a two-layer $MLP$ on top as a classifier. We tried both to freeze and not to freeze the previous neurons from *DAECL*.

As the final activation, we use Softmax, which is a sigmoid function for the binary cases. More formally, for $K$ classes, softmax corresponds to

$$\sigma(z_i) = \frac{e^{z_i}}{\sum_{j=1}^{K} e^{z_j}} \quad for\ i = 1, 2, \ldots, K \quad (7)$$

As a loss function for the classification, we minimize the use of Cross-Entropy:

$$\mathcal{L}_{FT} = -\sum_{k=1}^{K} y_k \log(p_k) \quad (8)$$

where $p_k$ is the predicted probability for the class $k$ and $y_k$ for the target. For binary classification datasets this can be further expanded as

$$\mathcal{L}_{FT} = -(y \log(p) + (1-y)\log(1-p)) \quad (9)$$

### 3.3.1 Joint

We wanted to check if we could benefit more when combining losses, i.e. by creating a joint loss based on the first and the second loss, (1 and 3), respectively.

$$\mathcal{L}_{Joint} = \mathcal{L}_{DAE} + \mathcal{L}_{CL} \qquad (10)$$

This training was obtained as a joint training of stages one and two. By adding the classification head, like in the previous section, for fine-tuning, we got the version we denote as *Joint* (see Table 3).

## 4 Experimental Setup

The datasets for the experiments, the two base models used and the metrics employed are detailed below.

### 4.1 Datasets

We have chosen several well-known datasets to carry out the experiments:

- Intent recognition on $SNIPS$ (SNIPS Natural Language Understanding benchmark) (Coucke et al., 2018). For this dataset, we found different versions, the first one with just 327 examples and 10 categories. This one was obtained from the huggingface library[3], which shows how this procedure performs on small datasets.

- The second version for $SNIPS$ from $Kaggle$[4] containing more samples and split into 7 classes that we call $SNIPS2$ is the most common one.

- The third one is commonly used for slot prediction (Qin et al., 2019), although here we only consider task intent recognition. We used $SST2$ and $SST5$ from (Socher et al., 2013) for classification containing many short text examples.

- We add $AGNews$[5] (Zhang et al., 2015) to our list, a medium size dataset that shows our method over long text.

- We complement the experiments with $IMDB$[6] (Maas et al., 2011), a big dataset

---

[3]https://huggingface.co/datasets/snips_built_in_intents
[4]https://www.kaggle.com/datasets/weipengfei/atis-snips
[5]https://huggingface.co/datasets/gimmaru/ag_news
[6]https://huggingface.co/datasets/imdb

with long inputs for binary classification in sentiment analysis. The length and size of this data made us consider only the $RoBERTa$ as the base model.

We used Hugging Face API to download all the datasets apart from the second version of $SNIPS$. In some cases, there was no validation dataset, so we used $20\%$ of the training set to create the validation dataset. There was an unlabelled test set in $SNIPS$, so we extracted another $10\%$ for testing. The unlabelled data was used for the first training phase, not for the other ones. The first and second phases did not use the test set. We selected the best model for each of them based on the results of the validation dataset. The test set was used at the end of phase 3.

### 4.2 Models and Training

We carried out experiments with two different models, a small model for short text $all\text{-}MiniLM\text{-}L12\text{-}v2$ (Wang et al., 2020)[7], more concretely, a version fine-tuned for sentence similarity and a medium size model $RoBERTa$-base (Liu et al., 2019)[8] which we abbreviate as $RoBERTa$. The ranges for the hyper-parameters below, as well as the values of the best accuracy, can be found in Appendix A.1.

We use Adam as the optimiser. We test different combinations of hyper-parameters, subject to the model size. We tried batch sizes from 2 to 128, whenever possible, based on the dataset and the base model. We tested the encoder with both *frozen* and *not frozen* weights - almost always getting better results when no freezing is in place. We tested two different truncations' lengths based either on the maximum length in the training dataset plus a $20\%$ or the default maximum length for the inputs in the base model. We tried to give more importance to one phase over the other by applying data augmentation in $CL$ or extending the number of epochs for the Autoencoder (Appendix A.1). We increased the relevance of the first phase by augmenting the data before applying random noise instead of changing the number of epochs. We tune the value of $ratio$, getting $0.6$ as the most common best value. To increase the relevance of the second phase, we increased the number of inputs by creating more pair combinations. Since we increased

---

[7]https://huggingface.co/sentence-transformers/all-MiniLM-L12-v2
[8]https://huggingface.co/roberta-base

| Dataset | 3-*Phase* | *Joint* | *FT* | S-P | EFL | CAE | Self-E | STC-DeBERTa | FTBERT |
|---------|-----------|---------|------|-----|-----|-----|--------|-------------|--------|
| *RoBERTa* | | | | | | | | | |
| SNIPS | **99.81** | 94.92 | 91.01 | 99.0 | - | 98.3 | - | - | - |
| SNIPS2 | 98.29 | 98.0 | 97.57 | **99.0** | - | 98.3 | - | - | - |
| SST2 | 95.07 | 93.12 | 90.28 | - | **96.9** | - | - | 94.78 | - |
| SST5 | **56.79** | 53.88 | 52.27 | - | - | - | 56.2 | - | - |
| AGNews | 95.08 | 94.82 | 92.47 | - | 86.1 | - | - | - | **95.20** |
| IMDB | **99.0** | 95.07 | 91.0 | - | 96.1 | - | - | - | - |
| *all-MiniLM-L12-v2* | | | | | | | | | |
| SNIPS | **100.00** | 91.98 | 92.89 | 99.0 | - | 98.3 | - | - | - |
| SNIPS2 | 98.57 | 98.57 | 93.86 | **99.0** | - | 98.3 | - | - | - |
| SST2 | 93.89 | 90.04 | 88.21 | - | **96.9** | - | - | 94.78 | - |
| SST5 | 54.77 | 52.25 | 49.24 | - | - | - | **56.2** | - | - |
| AGNews | 94.83 | 94.28 | 89.57 | - | 86.1 | - | - | - | **95.20** |

Table 3: Performance accuracy on different datasets using $RoBERTa$ and $all\text{-}MiniLM\text{-}L12\text{-}v2$ models in %. 3-*Phase* refers to our main 3 stages approach, while *Joint* denotes one whose loss is based on the combination of the first two losses, and *FT* corresponds to the fine-tuning. We also add some SOTA results from other papers: S-P denotes $Stack\text{-}Propagation$ (Qin et al., 2019), Self-E is used to denote (Sun et al., 2020b) (in this case we chose the value from $RoBERTa$-base for a fair comparison), CAE is used to detonate (Phuong et al., 2022), STC-DeBERTa refers to (Karl and Scherp, 2022), EFL points to (Wang et al., 2021b) (this one uses $RoBERTa$-Large), and FTBERT is (Sun et al., 2020a)

### 4.3 Metrics

We conducted experiments using the datasets previously presented in Section 4.1. We used the standard macro metrics for classification tasks, i.e., Accuracy, F1-score, Precision, Recall, and the confusion matrix. We present only the results from Accuracy in the main table to compare against other works. The results for other metrics can be found in Appendix A.2.

## 5 Results

We assess the performance of the three methods against the several prominent publicly available datasets. Thus, here we evaluate the 3-*Phase* procedure compared to *Joint* and *FT* approaches. We report the performance of these approaches in terms of accuracy in Table 3

We observe a noticeable improvement of at least 1% with 3-*Phase* as compared to the second best performing approach, *Joint*, in almost all the datasets. In $SNIPS2$ with $all\text{-}MiniLM\text{-}L12\text{-}v2$ we get the same values, while in $IMDB$ with $RoBERTa$ we get a gap of 4 point and 8 for

the $SNIPS$ dataset with $all\text{-}MiniLM\text{-}L12\text{-}v2$. We apply these three implementations to both base models, except $IMDB$, as the input length is too long for $all\text{-}MiniLM\text{-}L12\text{-}v2$. *FT* method on its own performs the worst concerning the other two counterparts for both the models tested. Eventually, we may conclude that the 3-*Phase* approach is generally the best one, followed by *Joint*, and as expected, *FT* provides the worst. We can also observe that *Joint* and *FT* tend to be close in datasets with short input, while $AGNews$ gets closer results for 3-*Phase* and Joint.

We did not have a bigger model for those datasets with long input, so we tried to compare against approaches with similar base models. We truncated the output for the datasets with the longest inputs, which may reflect smaller values in our case. Since the advent of (Sun et al., 2020a), several current techniques are based on combining different datasets in the first phase through multi-task learning and then fine-tuning each task in detail. Apart from (Sun et al., 2020a), this is the case for the prompt-base procedure from $EFL$ (Wang et al., 2021b) as well. Our method focuses on obtaining the best results for a single task and dataset. Several datasets to pre-train the model could be used as a phase before all the others. However, we

| Dataset | Base Model | 3-Phase | Joint | DAE+FT | CL+FT | Extra Imb. | No Imb. | FT |
|---------|-----------|---------|-------|--------|-------|-----------|---------|-----|
| SNIPS | $all\text{-}MiniLM\text{-}L12\text{-}v2$ | **100.00** | 91.98 | 98.05 | 99.81 | 99.92 | 99.88 | 92.89 |
| SNIPS2 | $all\text{-}MiniLM\text{-}L12\text{-}v2$ | **98.57** | **98.57** | 94.14 | 97.86 | 98.00 | 97.85 | 93.86 |
| SST2 | $RoBERTa$ | **95.07** | 93.12 | 83.72 | 94.72 | 94.50 | 94.04 | 90.28 |
| SST5 | $RoBERTa$ | **56.79** | 53.88 | 46.06 | 56.52 | 56.24 | 55.84 | 52.27 |
| AGNews | $RoBERTa$ | 95.08 | 94.82 | 91.53 | **95.24** | 95.01 | 94.26 | 92.47 |
| IMDB | $RoBERTa$ | **99.00** | 95.07 | 93.00 | 94.86 | 97.00 | 94.76 | 91.10 |

Table 4: Ablation results. As before, $3\text{-}Phase$, *Joint*, and *FT* correspond to the 3 stages approach, joint losses, and Fine-tuning, respectively. Here, *DAE+FT* denotes the denoising autoencoder together with fine-tuning, *CL+FT* denotes the contrastive Siamese training together with fine-tuning, *No Imb.* means $3\text{-}Phase$ but skipping the imbalance correction, and *Extra Imb.* refers to an increase of the imbalance correction to a $min_{ratio}$=1.5 and $max_{ratio}$=20.

doubt the possible advantages of this as the first and second phases would negatively affect this learning, and those techniques focused on training the classifier with several models would negatively affect the first two phases.

To complete the picture, $CAE$ (Phuong et al., 2022) is an architecture method, which is also independent of the pre-training practice. A self-explaining framework is proposed in (Sun et al., 2020b) as an architecture method that could be implemented on top of our approach.

### 5.1 Ablation study

We conducted ablation experiments on all the datasets, choosing the same hyper-parameters and base model as the best result for each one. The results can be seen in Table 4.

We start the table with the approaches mentioned: $3\text{-}Phase$ and *Joint*. We wanted to see if combining the first two phases could produce better results as those would be learned simultaneously. The results show that merging the two losses always leads to worse results, except for one of the datasets where they give the same value.

We start the ablations by considering only *DAE* right before fine-tuning, denoting it as *DAE+FT*. In this case, we assume that the fine-tuning will *carry* all the class information. One advantage of this outlook is that it still applies to models that employ a small labelled fraction of all the data (i.e., the unlabelled data represents the majority). The next column, *CL+FT*, replaces *DAE* with Contrastive Learning, concentrating the attention on the classes and not the data distribution. Considering only the classes and fine-tuning in *CL+FT*, we get better results than in *DAE+FT*, but still lower than the $3\text{-}Phase$ in almost all the datasets. Right after, we add two extreme cases of the imbalance correction,

where *Extra Imb.* increases the upper bound for the ratio and *No Imb.* excludes the imbalance method. Both cases generally produce lower accuracies than $3\text{-}Phase$, being *No Imb.* slightly lower. The last column corresponds to fine-tuning *FT*.

All these experiments proved that the proposed $3\text{-}Phase$ approach outperformed all the steps independently, on its own, or combined the Denoising Autoencoder and the Contrastive Learning as one loss.

## 6 Conclusion

The work presented here shows the advantages of fine-tuning a model in different phases with an imbalance correction, where each stage considers certain aspects, either as an adaptation to the text characteristics, the class differentiation, the imbalance, or the classification itself. We have shown that the proposed method can be equally or more effective than other methods explicitly created for a particular task, even if we do not use auxiliary datasets. Moreover, in all cases, it outperforms classical fine-tuning, thus proving that classical fine-tuning only partially exploits the potential of the datasets. Squeezing out all the juice from the data requires adapting to the data distribution and grouping the vector representations according to the task before the fine-tuning, which in our case is targeted towards classification.

## 7 Future work

The contrastive training phase benefits of data augmentation, i.e., we can increase the number of examples simply through combinatorics. However, this can lead to *space deformation* for small datasets, even with the imbalance correction, as fewer points are considered. Therefore, overfitting

occurs despite the combinatoric strategy. Another advantage of this phase is balancing the number of pairs with specific values. This practice allows us, for example, to increase the occurrence of the underrepresented class to make its cluster as well defined as those of the more represented categories (i.e. ones with more examples). This is a partial solution for the *imbalance problem*.

In the future, we want to address these problems. For the unbalanced class in the datasets, seek a solution to avoid overfitting to the under-represented classes and extend our work to support a few shot learning settings ($FSL$). To do so, we are going to analyze different data augmentation techniques. Among others, Variational Autoencoders. Recent approaches for text generation showed that hierarchical *VAE* models, such as stable diffusion models, may produce very accurate augmentation models. One way to investigate this is to convert the first phase into a *VAE* model, allowing us to generate more examples from underrepresented classes and generally employ them all in the $FSL$ setting.

Finally, we would like to combine our approach with other fine-tuning procedures, like prompt methods. Adding a prompt may help the model gain previous knowledge about the prompt structure instead of learning the prompt pattern simultaneously during the classification while fine-tuning.

## Limitations

This approach is hard to combine with other fine-tuning procedures, mainly those which combine different datasets and use the correlation between those datasets, since this one tries to extract and get as close as possible to the target dataset and task. The imbalance correction could be improved, restricting to cases where the text is short because it could be too noisy or choosing the tokens more elaborately and not just stop words. It would be necessary to do more experiments combined with other approaches, like the prompt base, to know if they benefit from each other or if they could have negative repercussions in the long term.

## Acknowledgments

The authors would like to thank the members of the AIApps Research Group in the Huawei Ireland and London Research Centers for their valuable discussion and comments. We especially want to thank Milan Redzic, Tri Kurniawan Wijaya and Jinhua Du for their help.

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

# A Appendix

## A.1 Hyper-parameters

This appendix section shows the final hyper-parameters from the best results in Table 3. The column at the end on the right contains the search space used for training. Some of the values were not used for training, either because of computational limitations or because they were not realistic for some datasets.

| Dataset Best Hyper-parameters | | | | | | | |
|---|---|---|---|---|---|---|---|
| $Parameter$ | SNIPS | SNIPS2 | SST2 | SST5 | AGNews | IMDB | $SearchSpace$ |
| $RoBERTa$ | | | | | | | |
| Learning rate $DAE$ | 5e-5 | 5e-5 | 1e-5 | 1e-5 | 2e-4 | 1e-5 | {1e-4, 2e-4, 5e-5, 1e-5, 1e-6} |
| Learning rate $CL$ | 5e-5 | 5e-5 | 1e-5 | 1e-5 | 2e-5 | 1e-5 | {1e-4, 2e-4, 5e-5, 1e-5, 1e-6} |
| Learning rate $FT$ | 1e-5 | 1e-5 | 1e-5 | 1e-5 | 2e-5 | 1e-5 | {1e-4, 1e-5, 2e-5, 1e-6} |
| Epochs $DAE$[*1] | 12 | 4 | 12 | 12 | 4 | 2 | {2, 3, 4, 9, 12} |
| Epochs $CL$ | 12 | 12 | 12 | 12 | 4 | 4 | {2, 3, 4, 9, 12} |
| Max Epochs $FT$[*2] | 70 | 70 | 70 | 70 | 70 | 50 | {20, 50, 70} |
| Batch size $DAE$ | 32 | 32 | 64 | 64 | 16 | 6 | {2,6,8,12,16,32,64,128} |
| Batch size $CL$ | 32 | 32 | 64 | 64 | 16 | 6 | {2,6,8,12,16,32,64,128} |
| Batch size $FT$ | 32 | 32 | 64 | 64 | 16 | 6 | {2,6,8,12,16,32,64,128} |
| Eps in $FT$[*3] | 2e-5 | 2e-5 | 2e-5 | 2e-5 | 2e-5 | 2e-5 | {2e-05} |
| Use Length[*4] | 48[*4] | 108[*4] | 512 | 512 | 512 | 500[*4] | {[*5], [*4], 512} |
| Freezing Encoder | False | False | False | False | True | False | {True, False} |
| Deleting ratio[*5] | 0.6 | 0.7 | 0.6 | 0.5 | 0.6 | 0.6 | {0.3, 0.4, 0.5, 0.6, 0.7} |
| $all\text{-}MiniLM\text{-}L12\text{-}v2$ | | | | | | | |
| Learning rate $DAE$ | 5e-5 | 1e-5 | 1e-5 | 5e-5 | 5e-5 | - | {1e-4, 2e-4, 5e-5, 1e-5, 1e-6} |
| Learning rate $CL$ | 5e-5 | 1e-5 | 1e-5 | 5e-5 | 5e-5 | - | {1e-4, 2e-4, 5e-5, 1e-5, 1e-6} |
| Learning rate $FT$ | 5e-5 | 1e-5 | 1e-5 | 1e-5 | 1e-5 | - | {1e-4, 1e-5, 2e-5, 1e-6} |
| Epochs $DAE$[*1] | 4 | 4 | 4 | 12 | 3 | - | {2, 3, 4, 9, 12} |
| Epochs $CL$ | 12 | 12 | 4 | 12 | 3 | - | {2, 3, 4, 9, 12} |
| Max Epochs $FT$[*2] | 70 | 70 | 70 | 70 | 70 | - | {20, 50, 70} |
| Batch size $DAE$ | 32 | 32 | 32 | 128 | 32 | - | {2,6,8,12,16,32,64,128} |
| Batch size $CL$ | 32 | 32 | 32 | 128 | 32 | - | {2,6,8,12,16,32,64,128} |
| Batch size $FT$ | 32 | 32 | 32 | 128 | 32 | - | {2,6,8,12,16,32,64,128} |
| Eps in $FT$[*3] | 2e-5 | 2e-5 | 2e-5 | 2e-5 | 2e-5 | - | {2e-05} |
| Use Length | [*5] | 108[*4] | 512 | 108[*4] | 512 | - | {[*5], [*4], 512} |
| Freezing Encoder | True | False | False | False | True | - | {True, False} |
| Deleting ratio[*5] | 0.6 | 0.3 | 0.6 | 0.7 | 0.6 | - | {0.3, 0.4, 0.5, 0.6, 0.7} |

Table 5: Hyper-parameters configurations and search space of the experiments.[*1] means these are not real epochs since the input data is not always the same. The data was masked on the fly; therefore, each epoch differs. [*2] We used a an early stopping approach for the $FT$ phase. [*3] We only consider the epsilon hyperparameter in the AdamW optimizer for $FT$. The other two phases use the default value from the Transformers library (1e-06). [*4] This hyper-parameter was estimated initially with the training dataset with a large margin. This was applied for datasets with very short sentences, like $SNIPS$. [*5] This hyper-parameter estimates the max length of the sequences using the $10\%$ of the examples. This estimation is multiplied by 1.2 and is added as the maximum size of the sequences for the embedding layers. The difference is that it was done on the fly and not preserved in this case.

## A.2 Other metrics

In this section of the appendix, we present the results obtained for metrics other than accuracy. More specifically, we present three tables: Precision and Recall in Table 6, and F1 (Table 7). These metrics show better the role played by the imbalance correction. The notation follows the Table 3.

| Dataset | Precision | | | Recall | | |
|---|---|---|---|---|---|---|
| | $3\text{-}Phase$ | $Joint$ | $FT$ | $3\text{-}Phase$ | $Joint$ | $FT$ |
| $RoBERTa$ | | | | | | |
| SNIPS | **99.81** | 94.94 | 91.04 | **99.81** | 94.99 | 91.03 |
| SNIPS2 | **98.26** | 97.96 | 97.50 | **98.30** | 98.09 | 97.72 |
| SST2 | **95.62** | 94.29 | 92.53 | **95.05** | 92.32 | 89.21 |
| SST5 | **55.51** | 53.23 | 49.50 | **53.71** | 51.18 | 49.02 |
| AGNews | **95.09** | 94.82 | 92.44 | **95.08** | 94.82 | 92.47 |
| IMDB | **98.08** | 95.56 | 91.83 | **100.0** | 96.74 | 91.03 |
| $all\text{-}MiniLM\text{-}L12\text{-}v2$ | | | | | | |
| SNIPS | **100.00** | 92.05 | 93.12 | **100.00** | 92.06 | 92.98 |
| SNIPS2 | 98.61 | **98.64** | 93.89 | **98.68** | 98.60 | 94.14 |
| SST2 | **93.92** | 92.32 | 88.79 | **94.89** | 91.24 | 88.13 |
| SST5 | 59.57 | **61.62** | 53.15 | **49.02** | 46.02 | 40.02 |
| AGNews | **94.84** | 94.28 | 89.57 | **94.83** | 94.28 | 89.57 |

Table 6: Precision and Recall values. The best values are shown in bold

| Dataset | $3\text{-}Phase$ | $Joint$ | $FT$ | S-P | EFL | CAE | FTBERT |
|---|---|---|---|---|---|---|---|
| $RoBERTa$ | | | | | | | |
| SNIPS | **99.81** | 94.95 | 91.03 | 97.0 | - | 97.0 | - |
| SNIPS2 | **98.28** | 98.01 | 97.57 | 97.0 | - | 97.0 | - |
| SST2 | **95.14** | 93.29 | 90.82 | - | - | - | - |
| SST5 | **54.59** | 52.18 | 49.24 | - | - | - | - |
| AGNews | 95.08 | 94.81 | 92.45 | - | 79.5 | - | **95.20** |
| IMDB | **99.03** | 95.10 | 91.43 | - | - | - | - |
| $all\text{-}MiniLM\text{-}L12\text{-}v2$ | | | | | | | |
| SNIPS | **100.00** | 92.0 | 92.81 | 97.0 | - | 97.0 | - |
| SNIPS2 | **98.63** | 98.60 | 93.90 | 97.0 | - | 97.0 | - |
| SST2 | **94.39** | 91.78 | 88.46 | - | - | - | - |
| SST5 | **53.78** | 52.69 | 45.66 | - | - | - | - |
| AGNews | 94.83 | 94.27 | 89.56 | - | 79.5 | - | **95.20** |

Table 7: F1 values for the best results.The best values are shown in bold

