# OpenReview forum: "Combining Denoising Autoencoders with Contrastive Learning to fine-tune Transformer Models"
_EMNLP/2023/Conference — EMNLP 2023 Main_

### Official Review · Reviewer_j83o · 2023-08-04

**Soundness:** 3

**Excitement:**

2: Mediocre: This paper makes marginal contributions (vs non-contemporaneous work), so I would rather not see it in the conference.

**Paper Topic And Main Contributions:**

This paper propose a three-phase fine-tuning approach to adapt a pre-trained base model to a supervised classification task, yielding more favourable results than classical fine-tuning, and propose an unbalance correction method.

**Reasons To Accept:**

The authors provide extensive experimental results on several datasets to demonstrate their 3-stage finetuning method's performance.
It empirically shows the gain from DAE pretraining and contrastive pretraining. The total setup is detailed, ensuring reproducibility. It may provide a good empirical example for the community.

**Reasons To Reject:**

Personally, I find myself not surprised enough by reading this paper. Adding a denoising pretraining phase will help, adding a contrastive pretraining phase will help, but it seems reasonable from my viewpoint. So it maybe somehow lacks novelty by just combining different existing techniques. I am willing to change my rating if I am persuaded that these are not simple A+B.

Another problem is that while these stages of pretraining help, what is the training cost? Are these costs meaningful for cost-performance trade-off?



**Reproducibility:**

4: Could mostly reproduce the results, but there may be some variation because of sample variance or minor variations in their interpretation of the protocol or method.

**Reviewer Confidence:**

3: Pretty sure, but there's a chance I missed something. Although I have a good feel for this area in general, I did not carefully check the paper's details, e.g., the math, experimental design, or novelty.

**Typos Grammar Style And Presentation Improvements:**

Maybe the results from Table 4 (Joint method) are misplaced?

I personally like the first paragraph of introduction, but the following writing can be improved to state clear what is your procedure and what are related work in intro.

---

> ### Author Rebuttal · Authors · 2023-08-28
>
> Dear reviewer, thank you for your valuable comments and helpful feedback, which will be considered for the revised version of the paper. Specifically, we have changed the misplaced results in Table 4 and will rewrite the introduction section for clarity, separating the motivation and the related work.
>
> The reviewer claims the work lacks some novelty by combining existing approaches. First, we would like to remark that this specific setup has not been built in the literature yet. Moreover, we conducted an extensive study showing why the presented model is the most natural way to combine these methods. We hypothesized that differencing the proposed fine-tuning phases and utilizing them in that specific order (unsupervised->un. correction+self-supervised->supervised) is most suitable for the data representation:
>
> 	1. On the whole, we considered that the DAE model should be in the first place because the model learns for each sample to be robust against noisy inputs (unsupervised approach).
>
> 	2. Subsequently, we decided to incorporate CL to learn the similarities between different samples in the same class and the dissimilarities with the other class examples (self-supervised approach). In addition, we proposed a novel unbalance correction to select more frequently the underrepresented classes while avoiding the repetition of inputs (un. correction).
>
> 	3. Finally, fine-tuning the model to the given dataset will adapt the representation for the predefined task (supervised approach).
>
> We showed through the experiments that combining all the joint losses is not as good as having this specific configuration. Furthermore, we agree with the reviewer regarding the increased training cost of using this fine-tuning configuration. There is indeed an increasing time using these two previous fine-tuning phases. Still, we have demonstrated that the performance couldn't be that high by only fine-tuning the model in the classical way for the classification tasks, so testing this configuration is very valuable as long as the task goal is to increase the performance of the classifiers. Similarly to large language models, this training time could be reduced using higher computational power when there is any time constraint.

---

### Official Review · Reviewer_Nkv3 · 2023-08-05

**Soundness:** 4

**Excitement:**

4: Strong: This paper deepens the understanding of some phenomenon or lowers the barriers to an existing research direction.

**Paper Topic And Main Contributions:**

The paper explores the problem of adjusting a (large) pretrained model to a new supervised classification task. Specifically, the authors propose a new three-staged fine-tuning algorithm: denoising autoencoder training, supervised contrastive learning, and finally fine-tuning.

Contributions:
* New fine-tuning algorithm outperforming the classical fine-tuning.
* New unbalance correction method (used by the authors during one of the fine-tuning stages).


**Questions For The Authors:**

Question A: Equation 3. Do I understand it right: it is just the modulus of the difference of two numbers?
Question B: I suggest using bold font in table 3 to highlight the best result values.
Question C: In table 4, column "Joint", rows "SST2" and "SST5": are the values correct? (why are the values not the same as in the table 3)
Question D: Why did you decide to perform no ablation studies for the IMDb dataset?

**Reasons To Accept:**

Reasons To Accept:
* New fine-tuning algorithm outperforming the classical fine-tuning.
* New unbalance correction method used by the authors during one of the fine-tuning stages.
* Careful explanation of the ideas behind each stage of the proposed algorithm: how it helps to better exploit the data during fune-tuning (why the algorithm outperforms the classical fine-tuning).
* Task-independence of the proposed approach (it can be used to fine-tune a base model for different tasks, not only classification).
* Extensive experiments with many models and datasets.
* Ablation study.

**Reasons To Reject:**

Aside from minor linguistic errors (which are not difficult to correct), I do not see much reason to reject the paper.

**Reproducibility:**

4: Could mostly reproduce the results, but there may be some variation because of sample variance or minor variations in their interpretation of the protocol or method.

**Reviewer Confidence:**

2: Willing to defend my evaluation, but it is fairly likely that I missed some details, didn't understand some central points, or can't be sure about the novelty of the work.

**Typos Grammar Style And Presentation Improvements:**

* The paper title differs from the one here in OpenReview (no "How to")
* Lines 98,99: "NLU", "FNN". Missing decryption on first use.
* Line 237. "Figure 1" splits on two lines. Use unbreakable space.
* Line 278 (eq 3). The symbol for the loss differs from that of eq. 1 (L vs \mathcal L)
* Eq. 4. Not quite clear what "x" means.
* Line 321. Use \min for "min" (\mathrm font should be here, not italic).
* Line 345 (and other places). "DAECL". Maybe I am wrong, but currently it looks like $DAECL$ (inline math). I believe just italic here would be fine.
* Eq. 8 and eq. 9: Top space between the equation and text is bigger than bottom space.
* Line 369. No need for "please".
* Line 380. Provide a link to Huggingface.
* Line 421. "we used" (past simple), "we test" (present simple).
* Line 444. Formatting of "1e-05" is not correct (excess spaces around minus sign).
* Line 456: Sometimes it "3 - Phases", sometimes "3 Phases", sometimes "3Phases" (different symbols for one notion)
* Line 576. I believe better say "where each stage".
* Line 572: I think that "hard" better suits here than "complicated".
* Line 594: "between" what?

---

> ### Author Rebuttal · Authors · 2023-08-28
>
> Dear reviewer, we would like to thank you for your valuable comments and the useful feedback which will be considered for the revised version of the paper.
>
> We have taken into consideration the reviewer's comments in the "Typos, Grammar, Style and Presentation Improvements" section with detailed feedback:
> 1. The Title has been corrected now matching the one from the submission.
> 2. The abbrevations have been definded at the first ocurrence.
> 3. The symbol "~" has been added before each "\ref".
> 4. The loss symbol has been standarized to "\mathcal{L}" in all the equations.
> 5. It has been clarified in the text that the domain of the variable "x" is any possible size of a category, meaning an integer value between the size of the smallest category and the biggest one.
> 6. Replaced "min" by "\min" in all the ocurrences.
> 7. The problem from "DAECL" may be related to formatting issues from the submission web site. We do not see the error in our version, neither can find it in the pdf from the rebuttal. We will ensure this does not happens in later versions, probably presenting the middle models names in italic.
> 8. We have deleted the empty line before any equation to correct the extra space.
> 9. "please" deleted.
> 10. Missing footnotes for the datasets have been added including "\footnote{\url{https://huggingface.co/datasets/snips_built_in_intents}}" for snips.
> 11. We are standarizing the text to present simple.
> 12. The format of "1e-5" and any other scientific notation have been corrected.
> 13. We have standarized to "3 - Phases" all the text.
> 14. "where each takes" has been replaced by "where each stage takes".
> 15. We replaced "complicated" with "hard" in line 572, as it suits best.
> 16. This has been clarified with "between those datasets".
> Apart from the errors pointed out by the reviewer, a detailed revision of the text will be included in the final version to solve other possible minor errors.
>
> Question A: Yes, the reviewer understood it correctly. We used an equation to explain it better instead of writing it in the text. The description of the loss function we used can be found in the SBERT library:
> https://www.sbert.net/docs/package_reference/losses.html?#cosinesimilarityloss
>
> Question B: The authors found this suggestion useful and have highlighted the best values in bold.
>
> Question C: As the reviewer pointed out, there were two typos when adding the values to the table. The first one: in the column Joint, the values between SST2 and SST5 had been interchanged, but now they have been corrected. The second one is the value "92.74", which should be "92.47" because of a typing error in which the order of the decimals was reversed.
>
> Question D: IMDB is the dataset that takes longer because of its size and the inputs' length, forcing us to use a small batch size. At first, we decided to perform the ablation experiments with smaller datasets, which could provide enough evidence for our claims. However, we have carried out the experiments in the IMDB dataset as well for completeness, and the following final results will be included in the last version of the paper (including typo correction):
>
> Dataset | Base Model | 3 Phases | Joint | DAE+FT |CL+FT |Extra Unb. |No Unb. |   FT
>
> SNIPS       allMiniLM     100.00     80.15     98.05      99.81      99.92        99.88      77.85
>
> SNIPS2     allMiniLM      98.57      96.13     94.14      97.86      98             97.85       93.74
>
> SST2         RoBERTa     95.07      92.47     83.37      94.72      93.80         93.80      90.28
>
> SST5          RoBERTa    56.79      53.88     46.06      55.70      55.79         55.84      52.27
>
> AGNEWS   RoBERTa    94.99      93.37     91.53      95.18      94.30         94.18      90.59
>
> IMDB          RoBERTa   93.39      95.07     90.63      93.51       93.74        93.19       87.87
>
>
> To facilitate the reproducibility of the experiments we will make the code publicy available, including data preprocessing, and the training loops to replicate the experiments.

---

### Official Review · Reviewer_oYVS · 2023-08-14

**Soundness:** 4

**Excitement:**

4: Strong: This paper deepens the understanding of some phenomenon or lowers the barriers to an existing research direction.

**Paper Topic And Main Contributions:**

The authors propose a new multi-phase approach for fine-tuning transformers to a specific dataset. The first phase adapts the pre-trained transformer to the goal dataset distribution through a DAE (unsupervised). The second phase applies a soft clustering through using a  siamese architecture with a cosine similarity loss using positive and negative examples. Moreover, during the second phase they also apply their novel approach to correct for the unbalanced class sizes in the data. Finally, after the first two stages, they apply the final fine-tuning to the goal dataset and show that the first two phases significantly improve the performance of the model on multiple public datasets.

**Reasons To Accept:**

The method was explained well and was easy to understand. Also, the 2 additional phases before fine-tuning make complete sense and is an intuitive, yet simple, approach to increase the performance of the pre-trained model after fine-tuning. Also, ablation studies were decent and proving that the phases add value to the final fine-tuning.

**Reasons To Reject:**

* Since the authors mention the unbalanced datasets multiple times and propose an approach to tackle it, "accuracy" was not the best metric to report, specifically since they mention they computed precision, recall, and F-1 score as well (or they could be added to the appendix).

* Moreover, although some hyper-params, e.g. value of min and max ratios, where thoroughly searched over, but some other important hyper-params, e.g. the learning rates or larger max number epochs, where not searched and experimented.

**Reproducibility:**

4: Could mostly reproduce the results, but there may be some variation because of sample variance or minor variations in their interpretation of the protocol or method.

**Reviewer Confidence:**

4: Quite sure. I tried to check the important points carefully. It's unlikely, though conceivable, that I missed something that should affect my ratings.

**Typos Grammar Style And Presentation Improvements:**

There are a couple minor errors and typos in the text. For example, in Table 4, row SST5, column Joint I do not think the accuracy is 92.74 while the others are under 60. Or in the same table in the caption mentions "man_ratio" whereas that should be "max_ratio".

---

> ### Author Rebuttal · Authors · 2023-08-28
>
> Dear reviewer, we would like to thank you for your valuable comments and helpful feedback, which will be considered for the revised version of the paper.
>
> We will address the minor errors and typos exposed for the final version. The ablation table contained two typos, one being the values exchange between SST2 and SST5 at the Joint column. This gave rise to the confusion where the value 92.74 stood out from the others.
>
> According to the reviewer's comments, we have simplified the experiments and computed Precision, Recall, F1, Accuracy and confusion matrices that will be included in the paper Appendix. Moreover, we will add some comparisons between these new metrics to facilitate the understanding of how the unbalance correction affects one of the datasets. We would like to explain that Accuracy was used to report the results since this is the standard metric for the different datasets used in previous works, so our performance can be compared to prior studies.
>
> We conducted an extensive initial grid search using a reduced version of the datasets and found the two best learning rates proposed. We decided to omit this typical hyperparameter search to focus the reader's attention on the main experiments. However, we will describe the process of this grid search for each dataset in the corresponding section of the paper, mentioning all the values tested. Furthermore, we will make the code publicly available for reproducing the whole process.
>
> Regarding the paper page limitation, the authors followed the paper submission guidelines. It seems that the "Limitations" section does not count as part of the paper (see https://2023.emnlp.org/calls/main_conference_papers/#:~:text=will%20not%20count%20towards%20the%20page%20limit): "This section will appear at the end of the paper, after the discussion/conclusions section and before the references, and will not count towards the page limit." Please let us know whether this information has changed, and we will process it accordingly.

---

### Meta-Review · Area_Chair_XRuy · 2023-09-19

**Recommendation:** 4

**Metareview:**

This paper proposes a multi-stage adaptation technique (unsupervised+contrastive+supervised) to improve quality of transfered models. All reviewers find the experiments sound and the technique well-explained. Only concern is raised by Reviewer j83o related to the novelty of the work as individual stages are not novel. I believe combining different techniques and demonstrating the benefit has great value for the community.

---

### Decision · Program_Chairs · 2023-10-07

**Decision:**

Accept-Main

**Comment:**

This paper proposes a multi-stage adaptation technique (unsupervised+contrastive+supervised) to improve quality of transfered models. All reviewers find the experiments sound and the technique well-explained. Only concern is raised by Reviewer j83o related to the novelty of the work as individual stages are not novel. I believe combining different techniques and demonstrating the benefit has great value for the community.